# Detection of Cognitive Load Modulation by EDA and HRV

**DOI:** 10.3390/s25082343

**Published:** 2025-04-08

**Authors:** Alexis Boffet, Laurent M. Arsac, Vincent Ibanez, Fabien Sauvet, Véronique Deschodt-Arsac

**Affiliations:** 1Laboratoire IMS, CNRS, UMR 5218, Université de Bordeaux, Talence, France; alexis.boffet@u-bordeaux.fr (A.B.); laurent.arsac@u-bordeaux.fr (L.M.A.); 2Thales AVS FRANCE SAS, Mérignac, France; vincent.ibanez@fr.thalesgroup.com; 3Institut de Recherche Biomédicale des Armées (IRBA), Brétigny sur Orge, France; fabien.sauvet@intradef.gouv.fr; 4URP 7330 VIFASOM, Université Paris Cité, Hôtel Dieu, Paris, France

**Keywords:** cognitive load, EDA, HRV, 2-back, NASA-TLX

## Abstract

Electrodermal activity (EDA) and heart rate variability (HRV) offer opportunities to grasp critical manifestations of the nervous autonomic system using low-intrusive sensing tools. A key question relies on the capacity to adequately process EDA and HRV signals to extract cognitive load markers, a multifaceted construct with intricate neural networks functioning, where emotions interfere with cognition. Here, 34 participants (20 males, 19.2 ± 1.3 years) were exposed to two-back mental tasking and watching emotionally charged images while recording EDA and HRV. HRV signals were processed using variable frequency complex demodulation (VFCDM) and wavelet packet transform (WPT) to provide high- and low-frequency (HF and LF) markers. Three methods were used to extract EDA indices: VFCDM (EDA_TVSYMP_), WPT (EDA_WPT_), and convex-optimization (EDA_CVX_). Cognitive load and emotion epochs were distinguished by significant differences in NASA-TLX scores, mental fatigue, and stress, on the one hand; and by EDA_CVX_ and, remarkably, EDA_TVSYMP_ and HF-HRV_VFCDM_ on the other hand. A linear mixed-effects model and stepwise backward selection procedure showed that these two markers were main predictors of the NASA-TLX score (cognitive load). The individual perception of cognitive load was finally discriminated by k-means clustering, showing three profiles of autonomic responses relying, respectively, on EDA_TVSYMP_, HF-HRV_VFCDM_, or a mix of these two markers. The existence of EDA-, HRV-, and EDA/HRV-derived profiles might explain why previous attempts that have predominantly employed a single biosignal often remained unconclusive in evaluating the perceived cognitive load, thereby demonstrating the added value of the present approach to monitor mental-related workload in human operators.

## 1. Introduction

Human monitoring during a variety of mental tasks has become popular for exploring human performance, well-being, and safety in different contexts. As technological environments become more demanding, and due to the inherent risk of placing human operators in increasingly complex tasks, there is a need to assess the perceived cognitive load to avoid exceeding the user’s abilities. In this context, neurophysiological research has provided a growing appreciation that a tight coupling exists between cortical activities in high-level cerebral assemblies sustaining cognitive functions and the activity of the autonomous nervous system, which is reflected in the control of end-effectors, e.g., the sudoral glands and the sinus node. This central and autonomous interplay offers a good opportunity to use minimally intrusive tools to process the signals emitted by these effectors, which could lead to an improved understanding of an individual’s resilience to the task. To obtain an overall picture, researchers have captured relevant signals in EDA and HRV sensing, as the activity of sudoral glands reflects sympathetic control and HRV contains critical information about autonomic sympathetic and parasympathetic functioning.

The main challenge that researchers are facing is to distinguish emotions from cognition in these end-effector signals. In fact, both cognitive functioning and emotions have been shown to interact with the activity of the autonomous nervous system [1,2]. Thus, although EDA and HRV represent signals of interest when it comes to using low-intrusive tools, in numerous cases, their respective time courses reflect the dynamics of an intricate cortical circuit. The prefrontal cortex (PFC) is a key structure for executive functions [3]. PFC exerts inhibitory control on a subcortical circuit (amygdala, hypothalamus, and brainstem) involved in emotional regulations. So, according to the neurovisceral integration model, the peripheral autonomic responses reflect the behavior of intricate cognitive and emotional responses [4].

EDA, the galvanic skin response that fluctuates over time, has been widely used to explore emotional and attentional states [5]. Several methods have been proposed to distinguish between tonic and phasic responses within the EDA signal [6,7], and, notably, Greco et al. [8] developed a convex optimization-based EDA method, providing a reliable tool to describe autonomic response when varying the affective level of a stimulus. Taking account of the sporadic nature of the sudomotor control, where bursts of neural activity are interspersed by latency phases, the analysis of phasic EDA is useful to draw a picture specially impacted by true periods of sudomotor activity, which has demonstrated particular value in recent years [9]. Posada et al. [10] proposed an index called the Time-Varying Sympathetic index (TVsymp), based on the complex frequency-variable demodulation (VFCDM) of EDA, about which, they demonstrated high sensitivity to distinguish marked autonomic responses to cold pressure, baroreflex unloading (head-up tilt), and cognitive interplay during the interference inhibition Stroop test. The later Stroop test result illustrates that (at least some) markers of EDA clearly reflect the autonomic response to a cognitive activity, with high specificity and sensitivity associated with the VFCDM-derived marker, TVsymp.

As an alternative approach, a recent application of machine-learning classification methods showed that TVSymp achieved superior scores in classifying emotions in terms of arousal, valence, and their respective levels [11]. Unfortunately, when it comes to associating EDA, even TVsymp, with the brain activity linked to perceived cognitive load and not emotions, experimental studies are scarce, and researchers acknowledge that this remains a challenge [12,13]. In this vein, Setz et al. [14] aimed to distinguish work-related stress by introducing EDA markers in statistical classification routines by varying the presence of known stressors, like temporal pressure and task difficulty, that exceed the operator’s ability. Although the approach is appealing, the detection of work-related stress rather than cognitive load was the main objective in their study. As summarized by other researchers, EDA markers might identify global work-related stress induced by an excessive cognitive load, but assessing cognitive load per se has not received clear evidence [15,16,17].

Critical information derived from direct brain imaging has helped us gain a more complete picture of individual perception of cognitive load, but brain mapping needs much more sophisticated and intrusive methods [18,19]. Rather, sensing heart rate might provide an additional support to exploring cognitive load. Heart–brain interplay through the neurovisceral integration model [20] suggests that vagally mediated responses could add value to cognitive exploration. Yet, HRV analyses failed to offer a clearer picture than the abovementioned EDA approaches. As cognitive load is a multifaceted construct that rarely occurs in isolation, one can hardly infer cognition from sudomotor and cardiac manifestations when analyzed individually. Another limiting aspect may rely on obvious variability in individual ANS activity [8]. One possible scenario could be that autonomic responses concern sudomotor control, cardiac control, or both in different proportions, so that they are dominantly mirrored in EDA or in HRV, which has encouraged experimental attempts combining different biosignals [21,22,23].

The present study, conducted in young and healthy participants, aimed at generating moderate levels of cognitive load and emotions during two distinctive epochs, while extracting autonomic markers through EDA and HRV analyses. We attempted to identify EDA and HRV markers that are relevant for cognitive load assessment specifically, and for that, we included a clustering analysis to shed light on emergent individual profiles.

## 2. Materials and Methods

### 2.1. Ethical Approval

The present study was approved by the Ethics Committee, the CERSTAPS (Approval No. IRB00012476-2023-27-03-241) and conformed to the standards set by the Declaration of Helsinki (2024 revised version).

### 2.2. Participants

Forty-seven healthy sport students (31 males, 18.9 ± 1.5 years) gave their written informed consent to participate in the experiment that was part of their academic curriculum and for which they received credits. Only 34 participants (20 males, 19.2 ± 1.3 years) were selected for analyses due to corrupted signals linked to technical problems and detected epochs of cardiac resonant during the acquisition of ECG signals. None of participants reported neurological and physiological disorders, nor did they report uncorrected vision problems. Participants were asked to avoid alcohol and caffeinated beverages for the 24 h preceding the experiment, and to abstain from heavy physical activity.

### 2.3. Experimental Procedure

The experiments were conducted in a quiet room with a controlled ambient temperature of 20–22 °C. Throughout the entire experimental procedure, participants were seated on a chair positioned at a visual distance of approximately 0.6 m from a 19-inch computer monitor. The participants were exposed to three consecutive conditions (Figure 1): Baseline consisted of quietly watching an animal documentary for 10 min; EMOT consisted of viewing a 10-min slideshow made of 40 emotionally labeled images from the IAPS (International Affective Picture System) database; and COG consisted of 10 min of 2-back tasking with prior (not analyzed) 2-min familiarization with the task.

### 2.4. Emotion (EMOT)

The task developed to induce various emotional responses contained images from the IAPS image bank [24] with normative ratings for valence and arousal levels of emotion [25], with a valence ranging from unpleasant (unhappy, bored, hopeless, etc.) to pleasant (happy, satisfied, hopeful, etc.) and an arousal ranging from not aroused (calm, relaxed, asleep, etc.) to highly aroused (excited, stimulated, wide awake, etc.). In the context of the present study, images were specifically selected and divided into 4 distinct groups by using a method inspired from Yun et al. [26]:

Step 1: Minimize the gender effect.

Images that had a mean valence rating difference between female and male participants of less than 1 and a mean arousal rating difference between female and male participants of less than 0.8 were selected. This resulted in 735 pictures.

Step 2: Build picture groups eliciting different emotions.

For this, pictures were filtered according to their valence and arousal values and according to their standard deviation for the respective category (std < 2 for valence, std < 2.23 for arousal).

Positive Valence, High Arousal (PVHA)—For PVHA, pictures with valence and arousal ratings greater than 6 were selected. This resulted in 41 pictures. Pictures with high standard deviation were excluded to result in a set of 27 pictures.Positive Valence, Low Arousal (PVLA)—For PVLA, pictures with a valence greater than 6 and an arousal state less than 4 were selected. This resulted in 46 pictures. Pictures with high standard deviation were excluded to result in a set of 29 pictures.Negative Valence, High Arousal (NVHA)—For NVHA, pictures with a valence lower than 4.3 and an arousal state greater than 6 were selected. This resulted in 54 pictures. Pictures with high standard deviation were excluded to result in a set of 30 pictures.Negative Valence, Low Arousal (NVLA)—For NVLA, pictures with a valence lower than 4.3 and an arousal state lower than 4 were selected. This resulted in 30 pictures. Pictures with high standard deviation were excluded to result in a set of 25 pictures.

Step 3: Final image selection.

For this, pictures containing explicit violence and sexually explicit pictures were excluded. Finally, 10 images were randomly selected in each group, resulting in the following:PVHA—images 1650, 4608, 4643, 4660, 5621, 5629, 8030, 8158, 8191, and 8206;PVLA—images 1333, 1419, 1670, 1812, 2035, 2358, 2392, 2500, 2515, and 2598;NVHA—images 1300, 1930, 3500, 3530, 6231, 6260, 6563, 8485, 9910, and 9630;NVLA—images 2039, 2271, 2280, 2312, 2410, 2491, 2520, 2752, 7013, and 7078.

Step 4: Design the slideshow.

Each of the groups of images obtained was divided into two blocks of 5 images. From these 8 blocks of images, a slideshow was constructed as follows: (PVHA-1; PVLA-1; NVHA-1; NVLA-1; PVHA-2; PVLA-2; NVHA-2; NVLA-2).

Each stimulus (image) was displayed at the center of the screen for 5 s, with an inter-stimulus interval of 10 s, during which a cross (+sign) was displayed on the center of the screen. Participants were instructed to stay focused on the screen during the entire slide show; they were also told that no action on their part is required during the entire task.

### 2.5. Cognitive Task (COG)

The 2-back version of n-back task, a task widely used to impose mental demand by involving working memory, was used in the present study [27]. During 2-back tasking (COG), the participant had to press “Enter” on the keyboard each time a letter was identical to the letter displayed 2 positions earlier. Each letter was pseudo-randomly displayed at the center of the screen for 500 ms, with an inter-trial interval of 2500 ms. The stimulus set comprised 15 letters: A, B, C, D, E, H, I, K, L, M, O, P, R, S, and T. Performance metrics (reaction time, omissions, success rate, and errors) were recorded during all the tasks. The reaction times obtained on the n-back test averaged 635 ms ± 176 ms, with an accuracy rate of 76.8%. This result highlights participant engagement, as confirmed by the cognitive load level expressed through the NASA-TLX scores (Table 1).

### 2.6. Stress, Mental Fatigue, and Cognitive Load

Stress, mental fatigue, and cognitive load were assessed before and after each condition (Baseline, EMOT, and COG). Slider scales from 1 to 100 without visible graduations were used to assess self-reported levels of stress and mental fatigue, where 1 means absence of and 100 means extreme perceived levels of stress and fatigue. To clarify how scoring works, definitions of stress and fatigue were priorly displayed using the following sentences: “Stress is a normal physiological response of the body to an external event or something perceived as threatening”; and “Mental fatigue refers to an inability to perform cognitive tasks and is distinct from fatigue caused by lack of sleep. For example, after prolonged hours of work or exams, one may experience difficulties in concentration and thinking”.

Participants completed a validated French version of the NASA-TLX [28] to evaluate the cognitive load thanks to the global score. The NASA-TLX evaluates six key dimensions: mental demand, physical demand, temporal demand, performance, effort, and frustration level [29]. We went through components of the global score to focus on mental demand and effort as dimensions specifically affected by cognitive tasking.

### 2.7. Electrodermal Activity

EDA was recorded (sampling frequency 1 kHz, 12 bits resolution) using the setup provided by ADInstruments (Dunedin, New Zealand), including two Ag-AgCl electrodes linked to the PowerLab 8/35 device and a fully isolated galvanic skin response amplifier with low voltage, 75 Hz AC excitation (FE116, ADInstruments). The electrodes were placed on distal phalanges of the index and middle fingers of the left hand. Raw EDA data were visually inspected for noise linked to unexpected motion-period identification principally linked to motion. Signals with more than 2% corrupted data were discarded.

For the main purpose of the present study (obtaining the finest detection of the cognitive load), the phasic component of EDA was extracted by using different methods based either on convex optimization (cvxEDA), variable frequency complex demodulation (VFCDM), or continuous wavelet (packet) transform (WPT). The cvxEDA algorithm [8] formulates EDA signal decomposition as an optimization problem to separate a slow varying tonic component and a more rapidly fluctuating phasic component. Here, we used the online available cvxEDA algorithm [30], with the following default parameters: α = 0.008, τ1 = 0.7 s, τ2 = 2 s, γ = 0.01, and sparse QR solver = “quadprog” (see [8] for more details). To avoid excessive computation time, EDA (1kHz) was downsampled at 2 Hz before deconvolution by cvx. The time-dependent amplitude of the phasic signal obtained was finally computed by using Hilbert transform, resulting in a sympathetic index called EDA_CVX_ here.

VFCDM was used in combination with Hilbert transform to extract the amplitude of the EDA signal in a specific bandwidth of the whole EDA signal. The main asset of VFCDM relies on the dynamical adjustment of the central frequency of main decomposed components. By adding those components demonstrating the greatest changes in amplitude when controlled sympathetic arousals are experimentally induced (0.08 to 0.24 Hz), Posada-Quintero et al. obtained a sympathetic index called TVsymp [10]. We obtained this index the same way here, starting with 2 Hz downsampled and high-pass filtered, (0.01 Hz) as recommended. For clarity and because there are many EDA and HRV indices computed in the present study, TVsymp is called EDA_TVSYMP_ here.

Wavelet packet transform (WPT) is widely used to extract time and frequency information simultaneously from physiological signals. WPT offers a better resolution for each computed frequency band than the often-used discrete wavelet transform (DWT) because, in DWT, the number of datasets is halved by the transform at each level [31]. We used level 3 of WPT (WPT3) on 1Hz resampled raw EDA signal, which provided 8 components partitioning the 0.5 Hz (Nyquist) bandwidth; and we added components 2, 3, and 4, which covered the frequency range of 0.0625–0.25 Hz, close to TVsymp. Hilbert transform provided the amplitude of this component, which served to obtain a sympathetic index called EDA_WPT3_ (see below) here.

### 2.8. Heart Rate Variability

HRV was obtained from successive interbeat time intervals identified in raw ECG signals by Rpeak-to-Rpeak (RR) distance. ECG was recorded by the PowerLab 8/35 (ADInstrument, Dunedin, New Zealand) at a sampling frequency of 1 kHz, with 50 Hz Notch filter, using three electrodes and a bio-amplifier (FE132, ADInstruments). The collected RR time series were exported to MATLAB 2021a (Matworks, Natick, MA, USA) for further analysis, using available functions and custom-designed routines. RR series were visually inspected for artifacts. Occasional ectopic beats (irregularity of the heart rhythm involving extra or skipped heartbeats, such as extrasystole and consecutive compensatory pause) were replaced by 2 Hz cubic-spline interpolated adjacent RR interval values. Signals with more than 2 percent of corrupted data were discarded.

To compare them with the previous literature, classic HRV indices were computed: in the temporal domain, RMSSD on RR series; in the frequency domain, on 4Hz-resampled RR series; and low-frequency (LF) and high-frequency (HF) power after obtaining power spectral density by discrete Fourier transform (DFT) and corresponding bandwidths of 0.04–0.15 Hz and 0.15–0.4 Hz.

For the purpose of the present study, we also applied VFCDM and WPT to the 1Hz-resampled RR series. Each method decomposes HRV into distinct frequency bands (components) within the range 0 to Nyquist frequency (0.5 Hz). Level 3 of WPT (WPT3), using Daubechies4 as a wavelet form, extracted eight 0.0625 Hz wide components with equal resolution. To obtain a sympathetic index, components 2 and 3 were added (0.0625–0.1875 Hz); and to obtain a parasympathetic index, components 4, 5, 6, and 7 were added (0.1875–0.4375 Hz) [32]. VFCDM provided eight components covering the range 0–0.5 Hz (Nyquist). Component 2, centered on 0.12 Hz, was used to obtain a sympathetic index, and components 3, 4, and 5, centered on 0.2, 0.28, and 0.36 Hz, respectively, were used to provide a parasympathetic index. For all computed components, Hilbert transform provided the time-varying amplitude of the component.

Finally, to obtain a set of variables able to describe the functioning of the autonomous nervous system during each of the 3 experimental conditions, the fluctuating amplitudes obtained by Hilbert transform for the above-described deconvolution and time-frequency analyses were averaged over the duration of each experimental condition: Baseline, EMOT, and COG conditions.

Hence, the statistical analysis was performed on the set of variables EDA_CVX_, EDA_TVSYMP_, EDA_WPT3_, LF_VFCDM_, HF_VFCDM_, LF_WPT3_, and HF_WPT3_; the classic variables RMSSD, LF_DFT_, and HF_DFT_; and the self-reported variables VAS Mental Fatigue, VAS Stress, NASA_TLX Global, NASA_TLX Mental Demand, and NASA_TLX Effort.

### 2.9. Statistical Analysis

Before testing the differences between each condition, the assumptions of normality and sphericity were checked using the Shapiro–Wilk normality test and Mauchly’s test of sphericity, respectively. In the case of a violation of the sphericity assumption, a Greenhouse–Geisser correction was applied.

If the assumptions were met, a one-way repeated-measures ANOVA was performed, followed by multiple pairwise paired *t*-tests. In the case of assumption violation, the alternative non-parametric Friedman test was used, followed by multiple pairwise paired Conover tests. All resulting *p*-values were then adjusted using the Bonferroni correction.

Moreover, a linear mixed-effects model was implemented using the lme4 package to assess the influence of physiological markers on cognitive load (NASA-TLX), with subject-specific random effects included to account for inter-individual variability.

The model initially incorporated the following predictors: EDA_CVX_, EDA_TVSYMP_, EDA_WPT_, HF_VFCDM_, and HF_WPT3_, obtained in each of the three conditions (Baseline, EMOT, and COG).NASA-TLX_Global~EDA_CVX_ + EDA_TVSYMP_ + EDA_WPT3_ + HF_VFCDM_ + HF_WPT3_ + 1|subject

Model convergence was assessed using the restricted maximum likelihood (REML) criterion. Multicollinearity was evaluated via the variance inflation factor (VIF), leading to the exclusion of HF_WPT3_ (VIF > 10) to prevent collinearity issues. A stepwise backward selection approach was applied to retain only significant predictors.

The final model reported fixed-effect estimates with corresponding standard errors (SE), t-values, 95% confidence intervals (CI), and *p*-values. The random effect variance for subjects and residual variance were also reported to assess individual variability in NASA-TLX scores. The significance threshold was set at *p* < 0.05.

To explore individual responses to cognitive demand, a k-means clustering analysis was performed on data collected during the COG condition, using the most relevant metrics identified from the linear mixed-effects model. The Elbow Method and the average silhouette width were applied to determine the optimal number of clusters.

All statistical analysis were performed under the RStudio software (Version 2023.12.1+402).

## 3. Results

### 3.1. Self-Reported Stress, Mental Fatigue, and Cognitive Load

During EMOT and COG, participants reported levels of mental fatigue and cognitive load above those reported during Baseline (Table 1).

Interestingly, NASA-TLX results showed that COG elicited higher cognitive load than EMOT. This holds for both subdimensions, mental demand and effort (Table 1), which strongly rely on cognitive load in our condition.

### 3.2. Physiological Indices

#### 3.2.1. Electrodermal Activity (EDA)

All indices derived from phasic EDA increased during COG when compared to Baseline and EMOT (Table 2), thus pointing to a heightened EDA response especially in COG condition.

#### 3.2.2. Heart Rate Variability (HRV)

As a main observation, not all HRV indices highlighted a specific response to COG; only HF-HRV markers showed a significative difference between COG and EMOT (Table 2). More explicitly, HF-HRV indices obtained by using VFCDM and WPT3 methods (HF_VFCDM_ and HF_WPT3_) showed lower values during COG, while RMSSD and DFT-derived indices that are also employed classically to describe the vagal behavior did not reach statistical significance (Table 2).

### 3.3. Linear Mixed-Effects Model (LMM)

Considering the NASA-TLX score to be a dependent variable, used here as a marker of cognitive load, the linear mixed-effects model included significant markers of COG: EDA_CVX_, EDA_TVSYMP_, EDA_WPT3_, HF_VFCDM_, HF_WPT3_, and random effects for subjects. The model showed good convergence, with a REML criterion value of 881.8. After verifying multicollinearity (VIF > 10), the predictor HF_WPT3_ was excluded from the model due to potential collinearity issues. A stepwise backward selection procedure was employed to obtain a parsimonious model by successively removing non-significant and poor predictors (EDA_CVX_ and EDA_WPT3_).

The fixed effects indicate that EDA_TVSYMP_ was positively associated with NASA-TLX (β = 13.80, SE = 2.00, t = 6.89 95% CI: [9.82, 17.71], *p* < 0.001), and HF_VFCDM_ showed a significant negative association with NASA-TLX (β = −6.24, SE = 2.23, t = −2.79 95% CI: [−10.63, −1.85], *p* = 0.003). The random effect variance for subjects was 100.2 (SD = 10.01), which suggests moderate inter-individual variability in NASA-TLX scores, with residual variance of 300.4 (SD = 17.33). The correlation between the fixed effects was low (0.350 between EDA_TVSYMP_ and HF_VFCDM_), indicating little multicollinearity between the predictors.

Taken together, these results suggest that both a high sympathetic activation reflected in EDA_TVSYMP_ and a low parasympathetic activity reflected in HF_VFCDM_ largely explain the individual level of perceived workload.

### 3.4. Analyses of Interindividual Responses

To further explore how the abovementioned relevant EDA and HF-HRV markers separately or collectively reflect the perceived cognitive load, we attempted to characterize individual responses, an approach also guided by the fact that each variable obviously spread out from its averaged value, reinforcing the idea of different typical links between cognitive load and specific marker(s). A k-means clustering analysis reveals that autonomic profiles might be apprehended through three clusters. The clustering explained a significant proportion of the total variance, as indicated by a ratio of between-cluster sum of squares (between_SS) to total sum of squares (total_SS) of 60.3%. The total within-cluster sum of squares (WCSS) across all clusters was 80.23, indicating tight clustering within groups. This value was substantially lower when using three clusters (80.23) compared to two clusters (108.61), indicating that the clustering solution with three clusters provides more compact and homogeneous groups. To validate the reliability of the three-cluster solution, the Elbow Method and the average silhouette width were applied, and both method confirmed the suitability of the three-cluster results.

Cluster 2 exhibits the highest variability (WCSS = 30.28), indicating that the points assigned to this cluster are more dispersed around their centroid compared to the other clusters.

Clusters 1 and 3 show lower variability (WCSS of 24.54 and 25.42, respectively), suggesting more compact groups compared to cluster 2. The autonomic profile of the participants grouped in the same cluster provided important information. Cluster 1 is composed of 39 individuals exhibiting a moderate negative EDA_TVSYMP_ values and high positive HF_VFCDM_ values. This cluster represents participants with elevated heart rate variability (HRV). Cluster 2 (24 individuals) points to both negative EDA_TVSYMP_ and HF_VFCDM_, with the strongest negative HF_VFCDM_ values. This cluster shows participants with reduced physiological variability. Cluster 3 (39 individuals) puts emphasis on the highest value of the EDA_TVSYMP_, instead, with a moderate negative value of HF_VFCDM_. Such results could indicate that neither EDA alone nor HRV alone is able to reflect the perceived cognitive load with acceptable reliability in a large population.

## 4. Discussion

Cognitive load and emotional responses are hard to distinguish at end-effector levels because of common neural networks modulating the autonomic nervous system. The main contribution of the present study is reliable information obtained by extracting autonomic markers in both EDA and HRV responses. In our conditions, relevant markers emerged from frequency-domain analyses, especially when using Variable Frequency Complex Demodulation of EDA and HRV signals. A clustering analysis capitalizing on statistically relevant markers shows that cognitive load is better reflected in levels of EDA_TVSYMP_ and HF-HRV_VFCDM_, respectively, a sympathetic marker of sudomotor responses and a vagal marker of heart rate modulations. The extraction of three clusters among our participants, based on these two markers after stepwise backward selection procedure, might explain why some previous studies using EDA or HRV alone have remained unconclusive about which autonomic signature can be associated with cognitive load [23,33]. Although physiological indices have been widely investigated in the laboratory to analyze responses to cognitive load, their application in real-world domains, such as aeronautics [34] and automotive environments [35], remains challenging. These limitations are primarily attributed to the difficulty of assessing workload in real time. To overcome these difficulties, combining EDA_TVSYMP_ and HF-HRV_VFCDM_ may constitute an interesting framework to obtain an individual profile of the perceived cognitive load during tasking, which has obvious applications for monitoring human operators.

As an experimental attempt here, young and healthy participants were exposed to a mental cognitive task (COG) and then to emotionally relevant pictures (EMOT). As the first step in the present study, self-reported levels of perceived cognitive load were significantly higher during COG than during EMOT conditions, as expected. This result holds for two subdimensions of NASA-TLX: mental fatigue and effort (Table 1). By comparison, stress did not differ between COG and EMOTION (Table 1). These results suggest that cognitive load and emotions were perceived distinctively by our participants, and they pave the way to interpreting this distinction in autonomic markers captured in EDA and HRV. It should be added that individual scores obtained in NASA-TLX, as well as in the stress test, varied substantially, although our participants had quite similar age and health status. NASA-TLX scores ranged from 46 (high load) to 81 (very high load), meaning that performing the same cognitive task (two-back) certainly requires different cognitive resources in an individual way. Such variability in psychophysiological responses to cognitive tasking has already been observed in the literature, and typically encourages a more detailed exploration of individual behavior [36].

Aligned with the distinction of self-reported cognitive load and stress after COG and EMOT, physiological markers derived from time-varying frequency analyses of EDA and HRV during the task exhibited different changes when COG is compared to EMOT. EDA_TVSYMP_, EDA_CVX_, and EDA_WPT3_ all indicated a more marked sympathetic increase during COG when compared to EMOT (Table 2). Given that previous usage of EDA to characterize the cognitive load in the literature has not provided such clear indications [15,16], the superiority of EDA markers extracted from the time-frequency domain analysis already suggested [10] could represent a first relevant step in addressing sympathetically driven responses to the perceived cognitive load in humans.

The HRV-derived behavior during COG and EMOT results also placed emphasis on time-frequency signal processing. Among the markers extracted from HRV, the distinction between COG and EMOT was highlighted when using VFCDM and WPT tools. The (more) classical Fourier-derived markers of HRV, as well as RMSSD, failed to distinguish COG from EMOT, thus reinforcing the relevance of extracting time-frequency markers. Because vagal and sympathetic heart rate modulations occur at different frequencies, the mode decomposition of HRV provides a more detailed picture of autonomic responses. In our conditions, greater relevance was observed in HF-HRV markers (HF_VFCDM_ and HF_WPT3_) than in LF-HRV markers. As the HF component accounts dominantly (although not exclusively) for vagal (parasympathetic) autonomic modulations of heart rate, it is concluded that n-back cognitive tasking, as used here, provides main adaptations in vagally mediated responses. In short, sympathetic responses derived from VFCDM applied to EDA in combination with concomitant vagal changes observed in HF-HRV_VFCDM_ provided a relevant picture to assess cognitive load in this study.

Although recent research also indicated that CVX and VFCDM are tools allowing EDA-derived indices to match with a change in cognitive load [15,16], the authors underlined obvious difficulty when it comes to further investigating individual responses [37]. For such a purpose, in our study, we introduced the additional sensing of HRV, which seemed to bring relevant additional information thanks to assessments of the cardiac vagal response. The marker HF-HRV, extracted from HRV by using VFCDM, could add substantial robustness when it comes to exploring further details, as observed here when applying stepwise backward selection to obtain a parsimonious model by successively removing non-significant and poor predictors of the NASA-TLX score. As an additional marker to EDA_TVSYMP_, HF_VFCDM_ emerged as a significant predictor of NASA scores, highlighting the relevance of these physiological measures for predicting performance outcomes. This result aligned with the importance of multi-sensor setups suggested by some authors to study physiological behavioral adaptations to complex tasks [23,33,38] and motivated a k-means clustering analysis that indeed revealed different autonomic profiles. A clustering solution validated by the Elbow Method provided three compact and homogeneous groups among our participants. Typical levels in HF-HRV_VFCDM_ and EDA_TVSYMP_ were associated with each group, pointing to people characterized by their high parasympathetic (HF-HRV) level and low sympathetic (EDA) level during COG, or by contrast, high sympathetic (EDA) together with a moderate vagal status; the third group demonstrated a weak autonomic status during COG characterized by low sympathetic (EDA) and very low parasympathetic (HF-HRV). To our knowledge, few studies have demonstrated the existence of clusters with distinct autonomic profiles in response to a cognitive task. By using a k-means clustering approach based on pupillary dilation dynamics, a recent study [39] identified two distinct autonomic response profiles. While this study provides valuable insights, it differs from ours in its methodological approach. Instead of conducting a preliminary analysis to identify the most sensitive markers for detecting cognitive load, it first differentiates clusters based on pupillary dynamics and subsequently analyzes the autonomic characteristics of each group.

This study is not without limitations. While the n-back task is validated and widely used in the literature, it is subject to certain biases, including inter-individual variability and the strategic approaches adopted by participants. These factors could contribute to the variability observed in autonomic responses in our results.

Although the model we used here provides valuable insights, it assumes a linear relationship between the predictors and the dependent variable. Given the complex dynamics of physiological signals, the relationship between variables could be more complex. A larger population would allow us to explore other nonlinear modeling approaches and possibly to generalize the model by performing a cross-validation procedure.

In conclusion, as mentioned above and as a main finding here, the existence of phenotypical autonomic behavior during a cognitive task underlines the real need to explore both EDA and HRV characteristics when using low-intrusive human monitoring. Extracting the autonomic manifestations at two end-effector levels could allow for a more robust interpretation of the perceived cognitive load, especially when VFCDM-derived markers are computed because they seem more prompt to reflect cognitive load than emotions during mental work conditions.

## Figures and Tables

**Figure 1 sensors-25-02343-f001:**
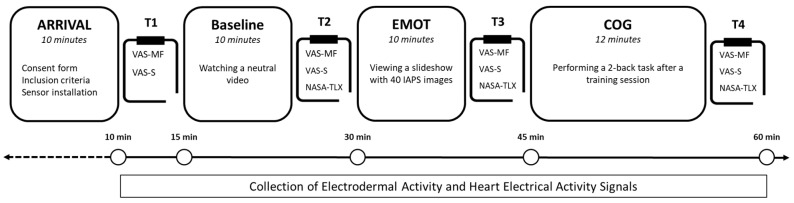
Experimental design. VAS-MF, Visual Analogic Scale of Mental Fatigue; VAS-S, Visual Analogic Scale of Stress; NASA-TLX, questionnaire of workload perceived.

**Table 1 sensors-25-02343-t001:** Comparisons of self-reported stress, mental fatigue, and cognitive load.

Metrics	Post Baseline T2	Post EMOT T3	Post COG T4
VAS Mental Fatigue	24 (9–37)	37 (18–54) *	49 (32–69) ***^,###^
VAS Stress	6 (0–11)	11 (0–18)	19 (5–34) ***^,#^
NASA_TLX Global	28 (18–35)	46 (29–67) ***	71 (59–82) ***^,###^
NASA_TLX Mental Demand	29 (20–40)	50 (30–80) ***	79 (70–90) ***^,###^
NASA_TLX Effort	19 (10–20)	37 (20–60) **	67 (50–90) ***^,###^

Values are displayed as median (quartile 1–quartile 3). *, **, and *** correspond to a difference with Baseline condition with a *p* < 0.05, *p* < 0.01, or *p* < 0.001, respectively. # and ### correspond to a difference with EMOT condition with a *p* < 0.05 or *p* < 0.001, respectively.

**Table 2 sensors-25-02343-t002:** Comparisons of physiological indices extracted from EDA and HRV.

Metrics	Units	Baseline	EMOT	COG
EDA_CVX_	n.u	1.05 ± 0.20	1.05 ± 0.20	1.26 ± 0.28 ***^,###^
EDA_TVSYMP_	n.u	0.80 ± 0.13	0.82 ± 0.12	1.04 ± 0.10 ***^,###^
EDA_WPT3_	n.u	0.94 ± 0.14	0.90 ±0.17	1.04 ± 0.11 **^,###^
RMSSD	ms^2^	45 ± 16	48 ± 18	48 ± 18
LF_DFT_	ms^2^	516 ± 308	693 ± 466	689 ± 468
HF_DFT_	ms^2^	310 ± 269	371 ± 284	329 ± 261
LF/HF_DFT_	n.u	2.5 ± 2.2	2.5 ± 1.7	2.74 ± 1.65
LF_VFCDM_	n.u	1.24 ± 0.04	1.22 ± 0.04	1.24 ± 0.03
HF_VFCDM_	n.u	1.28 ± 0.05	1.28 ± 0.06	1.24 ± 0.04 **^,##^
LF_WPT3_	n.u	1.21 ± 0.05	1.20 ± 0.05	1.22 ± 0.03
HF_WPT3_	n.u	1.27 ± 0.05	1.27 ± 0.05	1.23 ± 0.5 *^,#^

n.u: normalized units. Values are displayed as mean ± standard deviation. *, **, and *** correspond to a difference with Baseline condition with a *p* < 0.05, *p* < 0.01, or *p* < 0.001, respectively. #, ##, and ### correspond to a difference with EMOT condition with a *p* < 0.05, *p* < 0.01, or *p* < 0.001, respectively.

## Data Availability

The dataset generated and analyzed during the current study is available from the corresponding author upon reasonable request.

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
