# Peer review of "Detection of Cognitive Load Modulation by EDA and HRV"

_sensors, 2025, doi:10.3390/s25082343_

Round 1

Reviewer 1 Report

Comments and Suggestions for Authors

Τhis research attempts to describe and explain cognitive load modulation as a function  of Electrodermal activity (EDA) and heart rate variability (HRV). |The objective is to extract cognitive load markers, (a multifaceted construct with intricate neural networks functioning, where emotions interfere with cognition) using EDA and HRV signals.

The data collection was carried out via known valid procedures, while the data analysis was based on modeling the interindividual variability. Specifically, one-way repeated measures ANOVA (using three categories/profiles), and a the linear mixed-effects model were implemented. The three profiles (EDATVSYMP, on HF-HRVVFCDM or the mix of these) were extracted using the individual perception of cognitive load and the application of k-means clustering. The existence of EDA-, HRV- and EDA/HRV-derived profiles provides better explanation from other previous attempts which have used merely a single biosignal. This is the main finding, which even though not stunning, it adds to the literature, demonstrating the added value of the present approach to monitor mental-related workload in human operators.    

The paper is well written, and there is no observations on the main issues, however, a comment is that the phenomena under investigation are dynamic and linear models might not captured potential nonlinearity.

I would suggest adding a small special section addressing the limitations of the present research and the restrictions originating from methodological or other choices.

Author Response

Thank you very much for taking the time to review this manuscript.

Please find the detailed responses below and the corresponding revisions in track changes in the re-submitted file.

As two of the three reviewers suggest adding a small section addressing limitations of the present study, we find it interesting to include all reviewer’s remarks in this new section that you will find in the revised document.

Comments 1:

This research attempts to describe and explain cognitive load modulation as a function of Electrodermal activity (EDA) and heart rate variability (HRV). |The objective is to extract cognitive load markers, (a multifaceted construct with intricate neural networks functioning, where emotions interfere with cognition) using EDA and HRV signals.

The data collection was carried out via known valid procedures, while the data analysis was based on modeling the interindividual variability. Specifically, one-way repeated measures ANOVA (using three categories/profiles), and a the linear mixed-effects model were implemented. The three profiles (EDATVSYMP, on HF-HRVVFCDM or the mix of these) were extracted using the individual perception of cognitive load and the application of k-means clustering. The existence of EDA-, HRV- and EDA/HRV-derived profiles provides better explanation from other previous attempts which have used merely a single biosignal. This is the main finding, which even though not stunning, it adds to the literature, demonstrating the added value of the present approach to monitor mental-related workload in human operators.    

The paper is well written, and there is no observations on the main issues, however, a comment is that the phenomena under investigation are dynamic and linear models might not captured potential nonlinearity.

I would suggest adding a small special section addressing the limitations of the present research and the restrictions originating from methodological or other choices.

Response 1:

We agree that a nonlinear model enables a more refined approach, better suited to the complexity of physiological signals. Thus, this model could provide several advantages to explain cognitive load modulations based on the physiological indices, especially if the relationship between these variables is complex and did not follow a strictly linear structure. Although the linear mixed-effects model (employed in the present study) assumes a linear relationship between the predictors and the dependent variable, this approach was selected for two main reasons:

First, nonlinear models are more complex to interpret, particularly when integrating multiple physiological responses to explain an effect.

Second, the model was designed to build upon markers that had already been identified as relevant through prior analyses, ensuring that the model focused on established physiological indicators rather than introducing additional complexity.

Reviewer 2 Report

Comments and Suggestions for Authors

All comments are in attachment.

Comment on possible confounding as is possible with the study you have 

comment on the cognitive task used. Are there others and how does your task compare?

how did you account for individual baseline variability?

no cross-validation was undertaken for the datamining results?

Comments on the Quality of English Language

Need substantial improvement in grammar as shown in ms

Author Response

Thank you very much for taking the time to review this manuscript.

Please find the detailed responses below and the corresponding revisions in track changes in the re-submitted file.

As two of the three reviewers suggest adding a small section addressing limitations of the present study, we find it interesting to include all reviewer’s remarks in this new section that you will find in the revised document.

Comments 1:

Comment on possible confounding as is possible with the study you have comment on the cognitive task used. Are there others and how does your task compare?

Response 1:

We acknowledge that the n-back task may introduce several potential biases that could impact the results (Laine et al., 2018). These biases have been recently reviewed in a study examining their effects across various working memory tasks (Ritakallio et al., 2024):

  • Individual difference bias: Variations in basic cognitive abilities and attentional control among participants.
  • Strategy bias: The possibility that participants develop task-specific strategies.
  • Cognitive process engagement: The n-back task recruits multiple cognitive processes beyond working memory, such as attention and response inhibition.

To address these concerns, we have discussed these biases in the added "Limitations" section.

The selection of the n-back task in this study was driven by its extensive use in the literature as a standardized task imposing cognitive load on working memory. The n-back task was recently used to assess working memory in aircraft pilots, a context particularly relevant to our research due to its implications for transferring findings to real-world operational environments (Liang et al., 2024).

These considerations have now been incorporated into both the "Limitations" and "Perspectives" (Conclusion) sections.

Laine, M.; Fellman, D.; Waris, O.; Nyman, T.J. The Early Effects of External and Internal Strategies on Working Memory Updating Training. Sci Rep 2018, 8, 4045, doi:10.1038/s41598-018-22396-5.

Ritakallio, L.; Fellman, D.; Salmi, J.; Jylkkä, J.; Laine, M. Self-Reported Strategy Use in Working Memory Tasks. Sci Rep 2024, 14, 4893, doi:10.1038/s41598-024-54160-3.

Liang, Y.; Peng, X.; Meng, Y.; Liu, Y.; Zhu, Q.; Xu, Z.; Yang, J. Effect of Acute Stress on Working Memory in Pilots: Investigating the Modulatory Role of Memory Load. PLOS ONE 2024, 19, e0288221, doi:10.1371/journal.pone.0288221.

Comments 2:

How did you account for individual baseline variability?

Response 2:

In the present study, the data are paired, meaning that comparisons were conducted within- rather than between-subjects.

To account for individual variability, we incorporated a subject-specific random effect in the linear mixed model employed in this study. This approach allows to account for inter-individual differences in baseline measurements while appropriately managing repeated measures within subjects. These details are now added to the "Methods" section.

Additionally, we have clarified the statistical methodology of the clustering analysis in the ‘Methods’ section. Specifically, we employed k-means clustering, an unsupervised classification method that groups observations based on similarity. To ensure the robustness of cluster formation, we optimized the number of clusters (k) using the elbow method and the silhouette index.

Comments 3:

No cross-validation was undertaken for the datamining results?

Response 3:

We acknowledge that no cross-validation was performed. This decision stems from the initial objective of our study, which was not to generalize the model but rather to explore profiles within our dataset. Additionally, the small number of subjects, combined with intra- and inter-individual variability, posed challenges for applying a meaningful cross-validation procedure.

We have explicitly mentioned these limitations in a newly added 'Limitations' section.

Comments 4:

for this paragraph - discuss similar or different findings in the literature. what are you comparing to from previous work?

Lines 402-407: The extraction of three clusters among our participants, based on these two markers after stepwise backward selection procedure, might explain why some previous studies using EDA or HRV alone have remained unconclusive about which autonomic signature can be associated with cognitive load. Pointing to EDATVSYMP and HF-HRVVFCDM for that may constitute an interesting framework to get an individual profile of the perceived cognitive load during tasking, which has obvious applications for monitoring human operators.

Response 4:

Regarding the sentence: 'This might explain why some previous studies using EDA or HRV alone have remained inconclusive about which autonomic signature can be associated with cognitive load,'

→ Two references were added to highlight the interest in combining sensors (Ahmadi, 2024; Liu, 2023), emphasizing the limitations of EDA and HRV used independently."

Regarding the sentence: ‘Pointing to EDATVSYMP and HF-HRVVFCDM for that may constitute an interesting framework to get an individual profile of the perceived cognitive load during tasking, which has obvious applications for monitoring human operators.

→ a sentence with references was added at the beginning of the discussion

Masi, G.; Amprimo, G.; Ferraris, C.; Priano, L. Stress and Workload Assessment in Aviation—A Narrative Review. Sensors 2023, 23, 3556, doi:10.3390/s23073556.

Sriranga, A.K.; Lu, Q.; Birrell, S. A Systematic Review of In-Vehicle Physiological Indices and Sensor Technology for Driver Mental Workload Monitoring. Sensors 2023, 23, 2214, doi:10.3390/s23042214.

Comments 5

has this grouping been observed before or something similar - references?

462-466: “Typical levels in HF-HRVVFCDM and EDATVSYMP were associated with each group pointing to people characterized by their high parasympathetic (HF-HRV) level and low sympathetic (EDA) level during COG, or by contrast high sympathetic (EDA) together with a moderate vagal status; the third group demonstrated a weak autonomic status during COG characterized by low sympathetic (EDA) and very low parasympathetic (HF-HRV).”

Response 5:

→ a sentence with a reference was added at the beginning of the discussion (line 557)

Alshanskaia, E.I.; Portnova, G.V.; Liaukovich, K.; Martynova, O.V. Pupillometry and Autonomic Nervous System Responses to Cognitive Load and False Feedback: An Unsupervised Machine Learning Approach. Front. Neurosci. 2024, 18, doi:10.3389/fnins.2024.1445697

Reviewer 3 Report

Comments and Suggestions for Authors

The paper is very specific research paper on electrodermal activity (EDA) and heart rate variability (HRV) that can help grasping critical manifestations of the nervous autonomic system using low-intrusive sensing tools. Although it is related more to a Medical Research it has common avenues with the topics of Sensors Journal. It is well written and in tune with medical research taxonomy and methods

Author Response

Summary

Thank you very much for taking the time to review this manuscript.

Please find the detailed responses below and the corresponding revisions in track changes in the re-submitted file.

As two of the three reviewers suggest adding a small section addressing limitations of the present study, we find it interesting to include all reviewer’s remarks in this new section that you will find in the revised document.

Round 2

Reviewer 2 Report

Comments and Suggestions for Authors

The changes following the review are adequate